# Survival Motor Neuron Protein Participates in Mouse Germ Cell Development and Spermatogonium Maintenance

**DOI:** 10.3390/ijms21030794

**Published:** 2020-01-25

**Authors:** Wei-Fang Chang, Jie Xu, Tzu-Ying Lin, Jing Hsu, Hsiu-Mei Hsieh-Li, Yuh-Ming Hwu, Ji-Long Liu, Chung-Hao Lu, Li-Ying Sung

**Affiliations:** 1Institute of Biotechnology, National Taiwan University, Taipei 106, Taiwan; weifange@gmail.com (W.-F.C.); lintzuying0316@gmail.com (T.-Y.L.); hsujing912@gmail.com (J.H.); 2Center for Advanced Models for Translational Sciences and Therapeutics, University of Michigan Medical Center, Ann Arbor, MI 48109, USA; jiex@med.umich.edu; 3Department of Life Science, National Taiwan Normal University, Taipei 116, Taiwan; hmhsieh@ntnu.edu.tw; 4Department of Obstetrics and Gynecology, Mackay Memorial Hospital, Taipei 10449, Taiwan; hwu4416@yahoo.com.tw; 5Department of Obstetrics and Gynecology, Mackay Medical College, New Taipei City 252, Taiwan; 6Department of Obstetrics and Gynecology, Mackay Junior College of Medicine, Nursing, and Management, Taipei 11260, Taiwan; 7MRC Functional Genomics Unit, Department of Physiology, Anatomy and Genetics, University of Oxford, Oxford OX1 2JD, UK; jilong.liu@dpag.ox.ac.uk; 8School of Life Science and Technology, Shanghai Tech University, Shanghai 201210, China; 9Agricultural Biotechnology Research Center, Academia Sinica, Taipei 115, Taiwan; 10Animal Resource Center, National Taiwan University, Taipei 106, Taiwan

**Keywords:** SMN, gametogenesis, spermatogonium

## Abstract

The defective human survival motor neuron 1 (*SMN1)* gene leads to spinal muscular atrophy (SMA), the most common genetic cause of infant mortality. We previously reported that loss of SMN results in rapid differentiation of *Drosophila* germline stem cells and mouse embryonic stem cells (ESCs), indicating that SMN also plays important roles in germ cell development and stem cell biology. Here, we show that in healthy mice, SMN is highly expressed in the gonadal tissues, prepubertal spermatogonia, and adult spermatocytes, whereas low SMN expression is found in differentiated spermatid and sperm. In SMA-like mice, the growth of testis tissues is retarded, accompanied with gamete development abnormalities and loss of the spermatogonia-specific marker. Consistently, knockdown of *Smn1* in spermatogonial stem cells (SSCs) leads to a compromised regeneration capacity in vitro and in vivo in transplantation experiments. In SMA-like mice, apoptosis and accumulation of the R-loop structure were significantly elevated, indicating that SMN plays a critical role in the survival of male germ cells. The present work demonstrates that SMN, in addition to its critical roles in neuronal development, participates in mouse germ cell and spermatogonium maintenance.

## 1. Introduction

Recently, we and others reported that the levels of SMN protein correlate with the capacities of stem cell proliferation and differentiation in *Drosophila* [1] and mice [2]. In *Drosophila*, SMN is highly enriched in germline stem cells and a concentration gradient of SMN is observed [1,3]. In third instar larvae, overexpression of *Smn* in male germline stem cells results in testis growth and an increased number of spermatocytes while mutation of the *Smn* gene leads to defects in stem cell self-renewal and daughter cell differentiation. In mouse embryonic stem cells (ESCs), delayed cell growth and increased differentiation signals took place after knockdown of *Smn1* [2]. These findings suggest that SMN plays important roles in germ cell development and pluripotent stem cell maintenance, in addition to its well-known roles during neuronal development.

SMN is a major assembler of Sm proteins, PRMT5, and gemin proteins for the formation of the small nuclear ribonucleoprotein (snRNP) complex, which carries out pre-mRNA splicing and 3′ end processing [4,5,6]. SmB and SmD3, members of the Sm family, regulate the splicing event and are crucial for germ cell development in *Drosophila* [7]. PRMT5, the arginine methyltransferase that participates in the snRNP processing, is reportedly important for germline specification in both *Drosophila* and mouse [8,9]. SMN, as well as senataxin (SETX), interact with each other and are required for resolving R-loops, RNA:DNA hybrids that form over transcription pause sites, created by RNA polymerase II in transcription termination regions [10]. It has been reported that *Setx* knockout mice are defective in spermatogenesis, meiotic recombination, and meiotic sex chromosome inactivation [11]. These findings also suggest that SMN plays important roles in germ cell development and stem cell biology.

In the present work, we first documented the expression patterns of SMN in gonadal tissues in young and adult mice. The defects upon SMN deficiency in gametogenesis were further analyzed in an SMA-like mouse model (*Smn1^−/−^;SMN2^+^*), shRNA-based knockdown, and germ cell transplantation experiments. Our data show that SMN is not only involved in the development of germ cells but also participates in spermatogonia maintenance, providing new evidence for SMN’s participation in mammalian germ cell development.

## 2. Results

### 2.1. SMN Expression during Spermatogenesis

The protein levels of SMN in various adult mouse tissues were compared. Abundant SMN protein was expressed in the ovary, testis, and brain, compared to moderate levels in the spinal cord and low levels in the kidney (Appendix A). The SMN expression profile was further examined in male germ cells. The testis of 2- and 8-week-old male mice were used to isolate spermatogonia and spermatocyte based on the level of Thy-1 Cell Surface Antigen (THY1) expression by fluorescence-activated cell sorting (FACS). Three populations were identified: Spermatogonial stem cells (SSCs, THY1^+^), spermatocyte (THY1^−^FSC^Hi^), and sperm/spermatids/spermatocyte (THY1^−^FSC^Lo^) (Figure 1A). The cell specificity of FACS was confirmed by immunofluorescent staining. The spermatogonia marker PLZF (promyelocytic leukemia zinc-finger, also called zinc finger and BTB domain-containing 16, ZBTB16) in 8-week-old mice was expressed with a high percentage in sorted THY1^+^ spermatogonia cells (60%); whereas THY1^−^FSC^Hi^ cells showed a low percentage of PLZF signal (7.6%) and high percentage of meiotic marker SCP3 (synaptonemal complex protein 3) expression (91%) (Figure 1B) [12,13,14]. The expression levels of the transcripts of *Smn1* and germ cell markers were also determined in sorted populations by real-time PCR (Figure 1C). As expected, the spermatid-to-sperm stage marker acrosin prepropeptide variant 1 (*Acr1*) increased in 8-week-old adult THY1^−^FSC^Lo^ cells and the meiotic marker *Scp3* showed the highest expression level in THY1^−^FSC^Hi^ spermatocytes. In the THY1^+^ SSC population, *Smn1* showed a significantly high expression level in 2-week-old cells but was low in 8-week-old cells. In the THY1^−^FSC^Hi^ spermatocyte population, *Smn1* decreased in 2-week-old cells but increased in 8-week-old cells (Figure 1C). The different expression levels of *Smn1* transcripts between SSCs from 2- and 8-week-old mice are shown in Figure 1C, and the expression profile of SMN in spermatogonia was further examined with double staining of SMN and PLZF antibodies. For most paired spermatogonia (A-paired, Apr), which were connected to each other by a cytoplasmic bridge and represented the proliferating status [15,16], they expressed a high level of SMN and were compatible with the spermatocytes (Figure 1D, upper two panels). However, in A-single spermatogonia (As), which harbor potency for maintaining the spermatogonial stem cell population owing a relatively quiescent status [15,16], SMN decreased in the cytoplasm, in the testis of both prepubertal and adult males (Figure 1D, lower panels). No background signals were detected in the isotype control (Appendix A). On the other hand, the high expression level of SMN in proliferating spermatogonia could be correlated with the higher level of *Smn1* transcripts in THY1+ SSC from 2-week-old mice, as shown in Figure 1C, because most of the spermatogonia are still proliferating at this stage. Interestingly, SMN diminished in the elongated spermatid and sperm (Figure 1E). These results imply that SMN might be correlated with the differentiation and propagation of spermatogonia.

### 2.2. Developmental Delay in the Gonadal Tissues of SMA-Like Mice

A SMN-defective mouse model that mimics human SMA pathology has made great contributions to our understanding of SMN biology and SMA pathophysiology [17,18]. Here, we used an SMA-like mouse model [18] to study whether reduced SMN causes developmental defects in the germ cells of gonadal tissues. The mice used in this study approximately represent the intermediate severity type II SMA, with a survival time about 2 weeks, which was genotyped as *Smn1^−^^/−^;SMN2^+^* (SMA). Non-affected littermates (*Smn1^+^^/−^;SMN2^+^*, Ct) were used as control animals (Figure 2A). SMA-like mice exhibited a smaller body size, lower activity, and significantly decreased gonadosomatic index (GSI) of the testes/ovaries compared to the controls (Figure 2B,C). Immunohistochemistry (IHC) and quantitative assays of the testis revealed that fewer numbers of PLZF-positive spermatogonia cells present in the seminiferous tubules of SMA-like mice compared to those in controls (Figure 2D). Furthermore, fewer MKi67-positive spermatogonia-like cells surrounding the basement membrane of the seminiferous tubules decreased in SMA-like mice, indicating a reduced number of proliferating spermatogonia (Figure 2E). In females, a lower SMN signal (Figure 2F, upper panel) and higher frequency of atretic follicles were detected in SMA-like mice compared to control females (Figure 2F, lower panel). The decreased PLZF signal might link to the failure of spermatogonia maintenance in SMA-like mice. Thus, the function of spermatogonia was further examined by a transplant experiment.

### 2.3. Decreased Ability of Self-Renewal in SMN-Depleted SSCs

Considering the convenience of obtaining SMN-deficient spermatogonium, we performed knockdown of *Smn1* in magnetic-activated cell sorting (MACS) isolated spermatogonial stem cells (SSCs) of C57/BL6 mice to evaluate the effects of SMN depletion. Antibody-enriched THY1^+^ SSCs were used for lentiviral-based knockdown experiments. shSMN lentivirus-transduced SSCs displayed lower expression levels of SMN and PLZF protein in the Western blotting analysis (Figure 3A,B), consistent with the result shown in the SMA-like mouse model in Figure 2. THY1^−^ cells, which contain spermatocyte and somatic cells, did not express PLZF and showed lower expression after SMN knockdown (Figure 3A,B). The RNA levels of *Smn1*, sal-like protein 4 (*Sall4)*, and *Plzf* genes were also decreased following SMN knockdown in THY1^+^ cells (Figure 3C), which was consistent with the Western blotting analysis and immunofluorescent staining results (Figure 3D). Of note, apoptotic markers, such as transformation-related protein 53 (*Trp53)*, BCL2-binding component 3 (*Bbc3*, also known as *Puma*), and BCL2-associated X protein (*Bax*), had similar levels in the *Smn1* knockdown and control groups (Figure 3C).

Transplantation of SSCs into the seminiferous tubules of an infertile recipient animal is considered the gold standard assay to determine the germline competence of SSCs [19,20,21]. To investigate whether SMN contributes to SSC maintenance, we transplanted SMN-depleted SSCs isolated from β-Actin promoter-driven enhanced green fluorescent protein (EGFP) mice to the seminiferous tubules of a busulfan-treated recipient ICR male mice (Figure 4d for the transplant procedure). SMN deficiency completely abolished the homing ability of EGFP-SSCs transduced with shSMN, whereas control EGFP-SSCs transduced with shLacZ control shRNA could still survive, proliferate, and differentiate in the host seminiferous tubules (Figure 3E,F). The quantification of the control EGFP-SSC colony formation rate was close to a previous report [22] (Figure 3G). These results indicate that a reduction of SMN leads to defects in male germline development and spermatogonia maintenance.

### 2.4. Increased R-loop Signal Following SMN Depletion

Accumulation of the RNA:DNA hybrid structure, R-loop, was a recently identified phenomenon in SMN-deficient cells due to the increased transcription termination, which is concomitant with the induction of the p53 pathway and DNA damage response, manifesting as γ-H2A.X activation and U12-mediated intron retention [10,23,24,25]. Because SMN is a key component of R-loop resolving, which affects transcription termination and may further contribute to cell proliferation [10,26], we tested whether SMN depletion leads to an increase of the R-loop in SMA-like mice. The IHC staining (Figure 4A) against the R-loop structural marker, S9.6, showed no significant differences between SMA-like mice and littermate controls in the total number of S9.6-positive tubules (Figure 4B, upper panel). However, a significantly higher amount of the R-loop structure appeared in the testis of SMA-like mice, which contained over five S9.6-positive cells per tubule (Figure 4B, lower panel). Terminal deoxynucleotidyl transferase dUTP nick end labeling (TUNEL) staining, which detects DNA fragmentation, which is usually associated with apoptosis, was analyzed in the testis of SMA-like mice. We observed a significant increase in the percentage and average number of TUNEL-positive cells per tubule (Figure 4C,D). These data demonstrated that a deficiency of SMN would form abundant R-loop structures, which may contribute to the disruption of RNA transcription, and lead to massive cell death in testicular cells.

## 3. Discussion

It is well established that SMN plays key roles in neuron cell development. Genetic defects in the SMN gene is causative of the devastating neuromuscular disease SMA. In recent years, we and others have investigated the roles of SMN beyond the central neuron system. Accumulating evidence supports the notion that SMN participates in germline cell development from *Drosophila* to mammalian, as well as in the maintenance of pluripotent stem cells [1,2,27]. The present work strengthens this notion by demonstrating that SMN deficiency leads to retarded gonad development, and that loss of SMN causes defective SSC maintenance, in vitro and in vivo.

The present study documented the expression profiles of SMN in the testis and SSCs in mice. Higher levels of SMN were found in spermatocytes than in elongated spermatids or sperm, and surrounding sertoli cells (Figure 1). SMN protein forms a concentration gradient in the testis that is inversely proportional to the state of differentiation, which implies the possible function of SMN in germ cell development of regulating transcription activity or mRNA processing following the gradual termination of transcription during spermatogenesis [28]. This is further supported by the IHC result of the testis of young mice, where SMN colocalized with PLZF as an uneven pattern (Figure 1).

The decreased gonad mass and the reduction of MKi67 spermatogonia in the testis of SMA-like mice (Figure 2) suggest that SMN might participate in controlling cell proliferation in germ cells, which is consistent with previous studies in *Drosophila* and a different SMA-like mouse model (SmnC/C) with milder symptoms [1,27]. In the present work, we showed that PLZF, a DNA-binding transcriptional repressor that is crucial for regulating spermatogonia quiescence and proliferation [13,29], was decreased in the testis of SMA-like mice, indicating a loss of the spermatogia population (Figure 2) [30,31,32]. Interestingly, in the seminiferous tubules, the immunofluorescent experiments showed that spermatocytes express higher levels of SMN protein than spermatogonia, which warrants further research into the SMN biology in spermatocytes.

SALL4 is a major regulator in SSC and sustains the SSC pool by cooperation with PLZF [33,34,35]. Not surprisingly, levels of SALL4 were found to be decreased in SMN knockdown SSCs, similar to the changes of PLZF. Our results indicate that SMN deficiency leads to decreased levels of PLZF and SALL4, which directly affects male germ cell development (Figure 3). Importantly, SSCs with SMN depletion were not able to reform the population for spermatogenesis in the transplant experiments (Figure 3), suggesting that SMN is highly correlated with SSC proliferation and repopulation.

At the molecule level, our data indicate that the deficiency of SMN in germ cells results in elevated levels of unresolved RNA:DNA hybrids, the so-called R-loop, a structure that forms over transcription pause sites, contributes to genome instability, and induces repressive marks on chromatin [25,36]. SMN is the key player, which interacts with the RNA helicase SETX, affects transcription termination, and regulates polymerase II activity by resolving the R-loop structure [10,37]. Unresolved RNA:DNA hybrids, induced by SMN depletion, lead to transcriptional termination and consequently cell cycle arrest (Figure 4) [10,11]. It has also been reported that the unresolved R-loop structure due to SMN deficiency in neurons, and oxidative stress in muscles or cardiomyocytes lead to apoptosis [18,24,38,39,40,41]. Here, we provided additional evidence by showing that a loss of SMN in germ cells also induces programmed cell death in male testis and in developing oocytes in female SMA-like mice (Figure 2 and Figure 4). A similar phenomenon is also found in *Setx* knockout mice, including increased apoptosis and R-loop accumulation, which further supports the correlation of SMN and the R-loop structure [11].

We noticed that the gene/protein expression profiles revealed by different methods utilized in the present work, including FACS, real-time PCR, and IHC, did not always correlate with each other. This may be due to the differential sensitivities of the assays, but perhaps, more likely, it is caused by the heterogeneous nature of the cell populations. More accurate separation/characterization methods may improve this situation.

Taken together, our work demonstrates that SMN plays key roles in germ cell development in vitro and in vivo. These new roles, in addition to its well-known roles in neuronal development, may shed light on our understanding of this important protein.

## 4. Methods

### 4.1. Availability of Data and Materials

All animal maintenance, care, and procedures described within were reviewed and approved by the Institutional Animal Care and Use Committee of National Taiwan University (NTU) according to the protocol number (NTU-105-EL-68, 6th January 2016 approved, and NTU-107-EL-154, 11th January 2018 approved). All methods in the manuscript were performed in accordance with the relevant guidelines and regulations of NTU. Graphics and tables in the manuscript were prepared by the first and corresponding authors.

### 4.2. Animals and Tissue Sampling

Two-week-old and sexually mature 8- to 12-week-old ICR mice (BioLASCO, Taipei, Taiwan) were used for providing tissues, including ovary, testis, brain, spinal cord, and kidney, for Western blot analysis in Appendix A. Tissues were homogenized with a grinder, and proteins were extracted by RIPA lysis buffer (20-188, Merck Millipore, Darmstadt, Germany). Transgenic *Smn1*-deficient mice (SMA-like) were generated from Hsieh-Li’s laboratory [18] and bred from the Rodent Model Resource Center [C57BL/6/*Tg(SMN2) Hung Smn1^tm1 Hung^*] (National Laboratory Animal Center, NLAC, Taipei, Taiwan). Testes and ovaries of SMA-like mice were collected and fixed by 4% paraformaldehyde (PFA, 30525-89, Alfa Aesar, Ward Hill, MA, USA) at different ages. Gonadosomatic index (GSI) = (testis or ovary weight divided by body weight) × 100.

### 4.3. Fluorescence-Activated Cell Sorting (FACS)

Testis from 2- and 8-week-old C57BL/6JNarl mice (NLAC) were digested by Collagenase IV and Trypsin, incubated with rat anti-CD90.2 (THY1) antibody (553011, BD Biosciences, San Jose, CA, USA) or isotype rat IgG (559072, BD) control, and then labeled with anti-rat 488 secondary antibody (A21208). Cells were placed into an FACS tube for isolation with three populations, including THY1^+^ SSCs, THY1^−^FSC^Hi^ spermatocyte, and THY1^−^FSC^Lo^ sperm/spermatids/spermatocyte, by an FACS Aria III Sorter (BD). Dead cells stained with propidium iodide (PI, 1 μg/mL) signals were excluded.

### 4.4. Immunohistochemistry (IHC)

The testis and ovary tissues of mice were dissected and immersed in 4% PFA overnight at 4 °C, and then embedded into wax. Ovary and testis sections were stained by mouse monoclonal anti-SMN antibody (2.5 μg/mL, 610646, BD) overnight at 4 °C. The germ cell markers, rat anti-TRA98 (6 μg/mL, ab82527, Abcam, Boston, MA, USA), rabbit anti-PLZF (1 μg/mL, sc-22839, Santa Cruz Biotechnology, Inc., CA, USA), proliferation marker MKi67 (4 μg/mL, ab16667, Abcam), and R-loop detecting antibody S9.6 (2 μg/mL, ENH001, Kerafast, Inc. Boston, MA, USA) were stained in testis sections. Isotype IgG, including mouse (554121, BD), rabbit (550875, BD) and rat (559072, BD), were used as the negative control. Secondary antibodies (4 μg/mL) purchased from Thermo (Thermo Fisher Scientific, Waltham, MA, USA) were used as following: Alexa Fluor 488 goat anti-mouse IgG (A11029), Alexa Fluor 647 goat anti-mouse IgG (A21235), Alexa Fluor 488 donkey anti-Rat IgG (A21208), and Alexa Fluor 555 goat anti-rabbit IgG (A21429). For the PLZF- and MKi67-positive cell counting in the seminiferous tubules, the same staining procedure was performed and used the standard VECTASTAIN ABC system (PK6100, Vector Laboratories, Burlingame, CA, USA), and the AEC solution (00-1122, Thermo, Waltham, MA, USA) was used for conjugated horseradish peroxidase (HRP) antibody detection to visualize the staining outcome. Hematoxylin was used for the counter stain. The confirmation of the antibody specificity is shown in the Appendix A. For measurement of the spermatogonia in SMA-like mice, two to three non-serial sections containing 60–90 oval-shaped tubules ranging from 100 to 150 μm in diameter were counted for spermatogonia (PLZF-positive), surrounding proliferating spermatogonia-like cells (MKi67-positive), and R-loop structure (S9.6 positive). TUNEL assay was performed following the manufacturer’s instructions (In Situ Cell Death Detection Kit, 11684795910, Roche Applied Science, Mannhein, Germany) and TUNEL-positive tubules were quantified for around 90 oval-shaped tubules with a diameter ranging from 100 to 150 μm. Two to four mice were used for each genotype of mice for each assay.

### 4.5. Immunofluorescent Staining and Confocal Microscopy

Fixed cells were washed in D-PBS and permeabilized in 0.5% Triton X-100, followed by incubation in 2% BSA blocking solution. Samples were incubated overnight at 4 °C with primary antibodies, mouse-anti-SMN (0.8 μg/mL), anti-SCP3 (2.5 μg/mL, ab15093, Abcam), and rabbit anti-PLZF (1 μg/mL), and then washed with 0.25% PBS-Tween-20 and incubated with a secondary antibody conjugated to a fluorescence dye (Alexa Fluor 488 goat anti-mouse IgG, or Alexa Fluor 555 goat anti-rabbit IgG, A11029, and A21429, 4 μg/mL, Thermo) for 1 h. The cell nucleus was stained with 4′,6-diamidino-2-phenylindole (DAPI, 100 ng/mL, D9564, Sigma, St. Louis, MO, USA). Finally, fluorescent samples were mounted and observed with a confocal microscope (TCS SP5 II, Leica, Wetzlar, Germany).

### 4.6. Western Blotting

Cell lysates from tissues (30 μg/lane) were run on a 12% polyacrylamide gel and then transferred to a PVDF membrane. Following 1 h of incubation to block the non-specific binding with 5% non-fat dried milk in TBS (Bio Basic Inc., Markham, Ontario Canada), the first antibody was incubated overnight at 4 °C, including mouse anti-SMN antibody (0.03 μg/mL) and rabbit-anti-PLZF antibody (0.2 μg/mL). α-Tubulin (0.3 μg/mL, T5168, Sigma, St. Louis, MO, USA) was used as an internal control. After several washes containing 0.1% Tween-20 in TBS, the blot was incubated for 1 h with an HRP-conjugated goat anti-mouse and rabbit IgG secondary antibody (31430 and 31460, Thermo). The bound antibody was detected by a chemiluminescence detection system.

### 4.7. Spermatogonial Stem Cells’ (SSCs) Isolation and shRNA Knockdown

Six to seven days postpartum (dpp), the testes of C57BL/6JNarl mice were collected and dissected following the previous published study for immune-fluorescent staining, Western blotting analysis, and real-time PCR [42]. The birth date of mice was defined as 0 dpp. Briefly, Collagenase IV- and Trypsin-digested cells were incubated with anti-CD90.2 (THY1) antibody and then labeled with anti-biotin microbeads (130-090-485, Miltenyi Sigma Biotec, Bergisch Gladbach, Germany). Cells were placed into a collection tube of magnetic-activated cell sorting (MACS) to isolate THY1^+^ SSCs. Isolated THY1^+^ SSCs were cultured in the BSA medium following the recipe published previously [43] and THY1^−^ fraction containing somatic cells was cultured in 10% FBS/DMEM medium and both fractions were assessed for shRNA knockdown on the collecting day. Control shLacZ plasmid and shRN targeting the mouse *Smn1* gene were obtained from the RNAi core facility (shLacZ: TRCN0000072224, shSMN#1: TRCN0000072018, shSMN#2: TRCN0000072022). Plasmids were transfected into 293T cells to produce lentivirus for transduction into target SSCs and somatic cells. The RNA and protein of shRNA transduced SSCs and somatic cells were collected at the third day from the collecting day.

### 4.8. Germ Cell Transplantation

β-Actin promoter-driven EGFP (EGFP) 6–7 dpp ICR mice backcrossed with C57BL/6JNarl for 20 generations and were used for donor SSCs in the germ cell transplantation experiment [42,44]. The transplant procedure is briefly illustrated in Figure 4d. Six-week-old recipient ICR male mice were treated with 40 mg/kg busulfan (B2635, Sigma) and kept for 1 month to eliminate endogenous germ cells [45,46]. Following previously described procedures [42], the testicle THY1^+^ SSC cell suspension from 6–7 dpp EGFP mice (concentration of 2.5 × 10^6^ cells/mL) was mixed with Trypan Blue dye (T8154, Sigma), and approximately 20 μL of the cell suspension was injected through the efferent duct into the rete testis of the recipients. Two months later, the testes of the recipient animals were dissected for the detection of EGFP^+^ colonies. Colonies were defined as groups of transplanted cells that occupied at least 0.1 mm of the tubules.

### 4.9. Genotyping

Mouse tails were lysed in 1X PCR buffer supplemented with proteinase K (Novagen, Darmstadt, Germany) at 55 °C. Tail lysates were heat inactivated and PCR was performed using primers specific for mouse *SMN* and transgenic human *SMN2* [18]. Amplified PCR products were separated out by gel electrophoresis and detected by Safeview DNA staining dye (GeneMark, Taipei, Taiwan).

### 4.10. Gene Expression

Cell samples were collected, and total RNA was extracted by TRIzol^®^ reagent (15596-026, Thermo) and treated with RNase-free DNase I (M6101, Promega, Madison, WI, USA). Reverse transcription was performed using the SuperScript III First-Strand Synthesis System (18080-051, Thermo) by random hexamer primers. SYBR Green PCR master mix (KK4603, Kapa Biosystems, Inc., Woburn, MA, USA) and 200 nM of the forward/reverse primers in a final volume of 10 μL were mixed with 25 ng of template cDNA. Real-time PCR was performed using the Roche LightCycler (LC480, Roche, Mannhein, Germany), with the Ct value of *GAPDH* serving as the internal control. Primer sequences are listed in the Appendix A.

### 4.11. Statistics

All data were presented as means ± standard error of the mean (SEM) and unpaired comparison was analyzed using the Student’s t-test or one-way ANOVA with Dunnett’s multiple comparison test (GraphPad Software Inc., La Jolla, CA, USA). Significance was assumed at a *p*-value of 0.05.

## 5. Conclusions

In the present work, we demonstrated the expression profile of SMN in the gonadal tissues of young and adult mice. SMN deficiency impeded gametogenesis in an SMA-like mouse model (*Smn1^−/−^;SMN2^+^*), especially affecting spermatogonia maintenance based on shRNA-based knockdown and germ cell transplantation experiments. SMN loss also induced programmed cell death and the accumulation of the R-loop structure in the male testis of SMA-like mice. These data show that SMN not only participates in germline development but also sustains the survival of spermatogonium in mice.

## Figures and Tables

**Figure 1 ijms-21-00794-f001:**
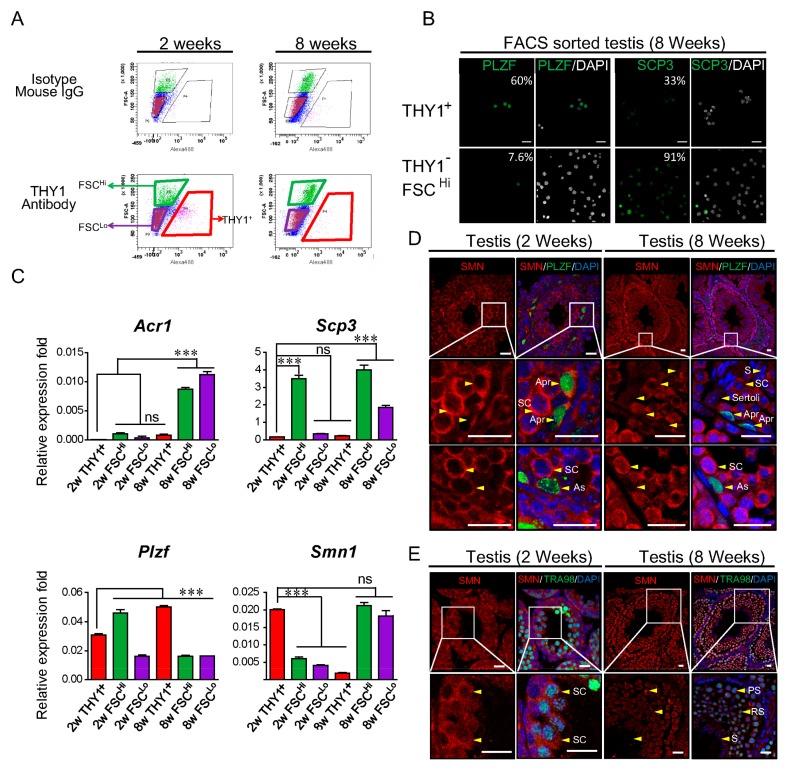
The distribution of survival motor neuron (SMN) in young and adult mouse testes. (**A**) Fluorescence-activated cell sorting (FACS) was used to characterize and sort testicular cells from 2- and 8-week-old mice. Based on the staining intensity of Thy-1 Cell Surface Antigen (THY1) conjugated with Alexa-488 fluorescent dye, three populations were identified: SSC (THY1^+^), spermatocyte (THY1^−^FSC^Hi^), and sperm/spermatids/spermatocyte (THY1^−^FSC^Lo^). Rat Immunoglobulin G (IgG) subsequently conjugated with 488 was used as the isotype control antibody. (**B**) Sorted THY1^+^ cells from 8-week-old mice express a high percentage (60%) of promyelocytic leukemia zinc-finger (PLZF) (green, left panel), whereas THY1^−^FSC^Hi^ cells shows a low percentage of PLZF signal (7.6%). The meiotic marker synaptonemal complex protein 3 (SCP3) was used to characterize the THY1^−^FSC^Hi^ population, which mostly contains spermatocyte. The percentages of markers in different populations are indicated. (**C**) Determination of the expression level of *Smn1* and germ cell markers in the sorted population. In the THY1^+^ SSC population, *Smn1* showed a significantly higher expression level in 2-week-old cells (2w THY1^+^) but decreased to a lower level in the THY1^−^FSC^Hi^ spermatocyte population (2w THY1^−^FSC^Hi^). In 8-week-old THY1^+^ SSC cells (8w THY1^+^), the expression level of *Smn1* showed no difference compared to the spermatocyte population of 2-week-old cells but increased significantly in the THY1^−^FSC^Hi^ spermatocyte population (8w THY1^−^FSC^Hi^). The transcript expression level of the spermatogonia-specific marker *Plzf*, sperm marker *Acr1*, and meiotic marker *Scp3* showed a cell-type-dependent manner. The error bar indicates technical repeats. The expression level of each group was compared with the 2w THY1^+^ group. *Indicates significance, *p* < 0.05; **, *p* < 0.005. (**D**) Putative proliferating A-paired spermatogonia (Apr, arrowhead, upper, and middle panel) expresses the abundant amount of SMN in the testis of both 2- and 8-week-old mice, whereas undifferentiated A-single spermatogonia (As, arrowhead, lower panels) expresses PLZF (green) and possesses a lower level of SMN (red) compared with spermatocyte (SC). Sertoli cells (arrow indicated) also express a lower amount of SMN protein. (**E**) Validation of germ cell marker TRA98 expression in spermatocyte. SMN expressed abundantly in pachytene (PS)-stage spermatocytes colocalized with TRA98 (green); the expression decreases in round spermatocyte (RS), elongated spermatocyte (ES), and is the lowest in spermatozoa (S) in 8-week-old testis. The isotype negative control used for the double staining of antibodies is shown in Appendix A. The scale bar in this figure represents 20 μm.

**Figure 2 ijms-21-00794-f002:**
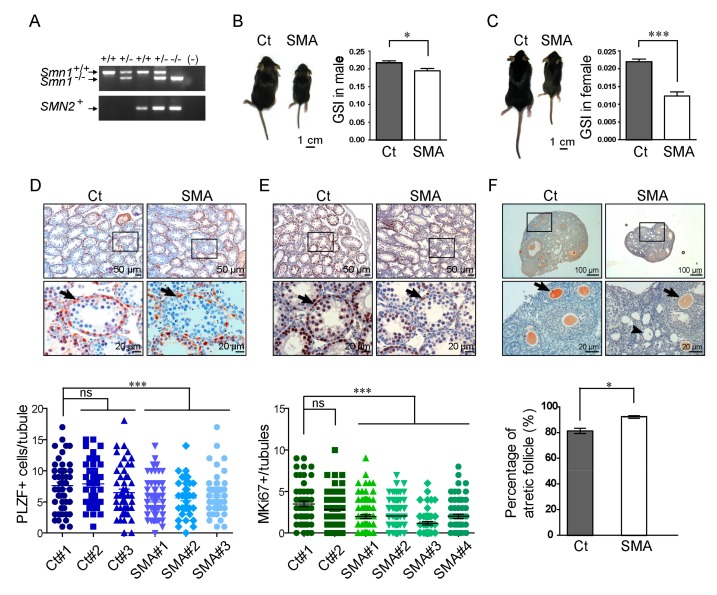
Testes and ovaries in SMA-like mice are abnormal. (**A**) Genotype of wildtype *Smn1^+/^^+^;SMN2^−^, Smn1^+/^^−^;SMN2^+^* (littermate control, Ct), and *Smn1^−/−^;SMN2^+^* (SMA-like) mice are confirmed by PCR. (-) indicates the negative control without a template. (**B**) Male SMA-like mice (indicated as SMA) with a phenotype has a smaller body size and significantly lower GSI than their heterozygote littermates (*Smn1^+/−^SMN2^+^*, indicated as Ct) at the age of 2 weeks (*n* = 4 Ct and 7 SMA mice). (**C**) The body size of female SMA-like mice (left panel), and the GSI of ovaries at 8 weeks after birth (right panel, *n* = 3 Ct and 4 SMA mice). (**D**) Reduction of PLZF-expressing germ cells (red color, arrows in larger magnification field) is shown in SMA-like mice but not in their heterozygote littermates after IHC analysis. Lower panel indicates its quantification. Each spot indicates the number of PLZF-expressing cells in each seminiferous tubule. Seminiferous tubules ranging from 100 to 150 μm are used for counting (*n* = 3 mice per genotype). (**E**) The expression of the proliferation marker, MKi67 (red color, arrow), is weaker in surrounding spermatogonia cells on the basal membrane of SMA-like mice than heterozygote controls. Hematoxylene was used as a counterstain (blue) (*n* = 2 for Ct and 4 for SMA). The negative control of IHC is shown in Appendix A. (**F**) IHC staining demonstrates low SMN protein levels (red color, arrow) in the immature oocytes in SMA-like mice. The percentage of atretic follicles (atresia) was quantified (lower panel). * Indicates a statistically significant difference, *p* < 0.05. ***, *p* < 0.001.

**Figure 3 ijms-21-00794-f003:**
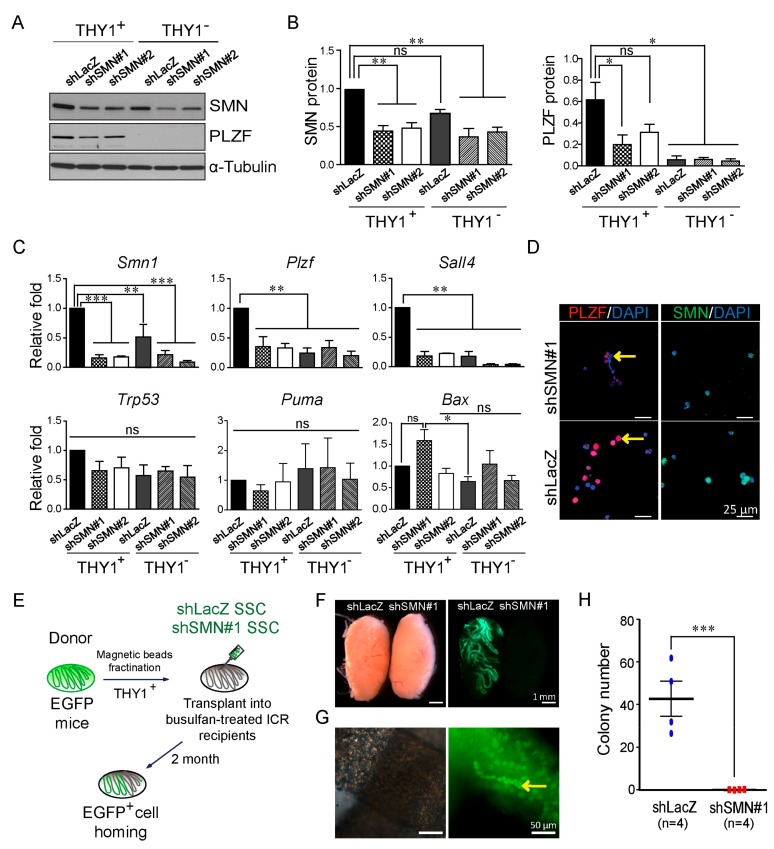
Spermatogonial stem cells (SSCs) show decreased PLZF expression and lost the ability of self-renewal after SMN knockdown. (A) Western blot analysis and its quantification (B) demonstrated decreased PLZF protein following lentiviral shSMN transduction in SSCs enriched by MACS isolated from 6–7 days postpartum (dpp) testes of C57BL/6JNarl mice (shSMN#1 and shSMN#2). Compared with the control shLacZ group, a significant difference is shown. α-Tubulin is used as the internal control of the Western blot. The original Western blot is shown in Appendix A (*n* = 2). (C) Real-time PCR analysis of SMN-depleted SSCs. *Smn1*, spermatogonia makers, *Plzf* and *Sall4,* and apoptotic markers *Trp53, Puma,* and *Bax* were analyzed **(*n*** = 3). (D) Immunofluorescent staining demonstrates SSCs express less PLZF in the SMN knockdown group (shSMN#1, red color, arrow indicated) compared to the control shLacZ group. 4′,6-diamidino-2-phenylindole (DAPI) is used as the counterstain (blue). (E) Schematic illustration of the process of SSC transplantation. (F) SSCs carrying enhanced green fluorescent protein (EGFP) transplanted into a recipient testis shows colonization in the shLacZ group (left testis) but not in SMN-depleted SSCs (shSMN#1, right testis). (G) Aligned EGFP-positive cells in the shLacZ control group represent the homing ability and normal proliferation (arrow). (H) Quantification of the SSC homing efficiency. Each spot indicates one replicate of the transplanted mouse. *Indicates significance, *p* < 0.05; **, *p* < 0.005; ***, *p* < 0.001; ns indicates no significance.

**Figure 4 ijms-21-00794-f004:**
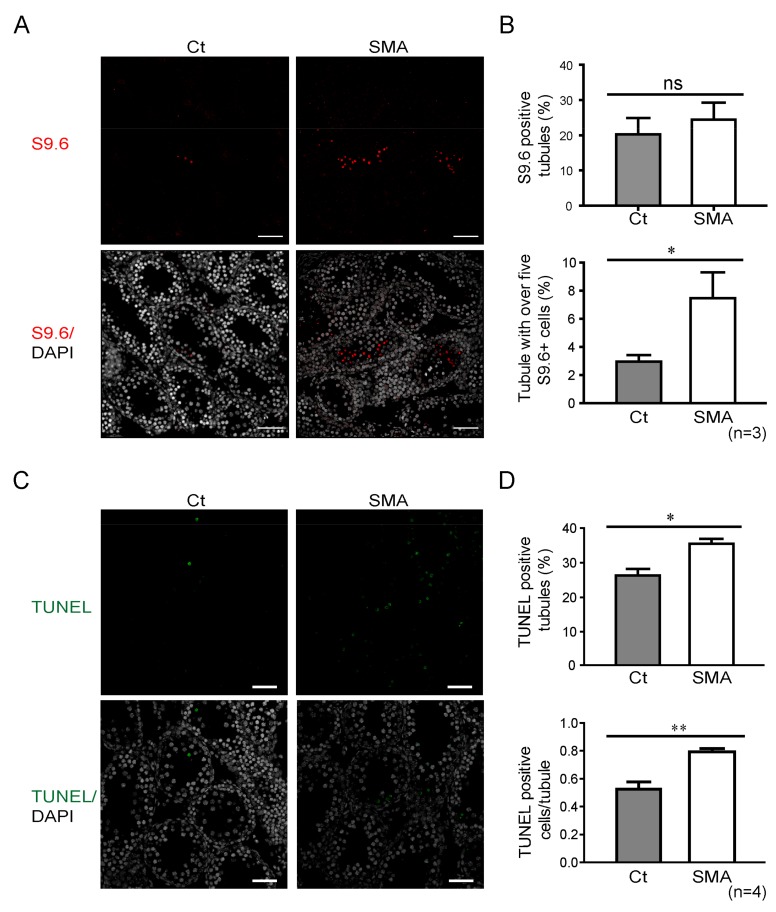
Detection of RNA:DNA hybrid structure in SMA-like mice. (**A**) Staining result of the RNA:DNA hybrid structure, R-loop, by the detecting antibody S9.6 (red) and DAPI (white) in SMA-like mice (SMA) and littermate controls (Ct) (scale bar = 50 μm). (**B**) Upper panel: Percentage of S9.6-positive seminiferous tubule of SMA-like mice (SMA, *n* = 3) and control littermates (Ct, *n* = 3 mice per genotype). Lower panel: Quantification of tubules containing over five S9.6-positive cells. (**C**) TUNEL assay of SMA-like mice (SMA) showing a higher amount of positive signal (green). Nuclei were stained with DAPI (white) (scale bar = 50 μm). (**D**) Percentage of seminiferous tubules with at least one TUNEL-positive cell (upper panel) and average number of TUNEL-positive cells per seminiferous tubule (lower panel). Over 90 tubules/mouse were counted for S9.6 and TUNEL. *Indicates significance, *p* < 0.05; **, *p* < 0.005.

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
