# Peer review of "Survival Motor Neuron Protein Participates in Mouse Germ Cell Development and Spermatogonium Maintenance"

_ijms, 2020, doi:10.3390/ijms21030794_

Round 1

Reviewer 1 Report

The study by Chang et al., reports defective human survival motor neuron 1 (SMN1) presence and role in mouse testis. Although the study is novel and interesting some improvemnts are suggested.

1. English needs special attention,
2. The study is overloaded with analysis and results
3. In reference to no.2 methodological enough details are not provided
4. In the abstract the information is not compatible as they are in conclusion (where it is provided in more detail)
5. graphs/microphotographs presented results are too small

Author Response

Response to Reviewer 1 Comments

The study by Chang et al., reports defective human survival motor neuron 1 (SMN1) presence and role in mouse testis. Although the study is novel and interesting some improvemnts are suggested.

Point 1: English needs special attention, 

Response 1: We have revised accordingly.

Point 2: The study is overloaded with analysis and results

Response 2: The present work is the first to systematically investigate SMN in germ cell development. Historically SMN is primarily studied in the context of neurobiology. As a result this manuscript is loaded with many analysis and results. We appreciate your understanding.

Point 3: In reference to no.2 methodological enough details are not provided

Response 3: Thank you for pointing this out. We have provided more details and corrected some typos in the revision.

Line 289 4.2. Animals and tissue sampling Two weeks old and sexually mature 8-12 week ICR mice (BioLASCO, Taipei, Taiwan) were used for providing tissues including ovary, testis, brain, spinal cord and kidney for western blot analysis in supplementary data. Tissues were homogenized with grinder and proteins were extracted by RIPA lysis buffer (20-188, Merck Millipore, Darmstadt, Germany).

Point 4: In the abstract the information is not compatible as they are in conclusion (where it is provided in more detail)

Response 4: We have modified our abstract in this revision.

Point 5: graphs/microphotographs presented results are too small

Response 5: Original size figure files are provided and the font size of figures was enlarged in this revision.

Reviewer 2 Report

In the present work, the authors studied the expression and the role of SMN in mouse testis. To this aim they focused on analysis of testis at two different ages, 2 and 8 weeks , and by using SMA-like mouse model (Smn1-/-;SMN2+), shRNA-based knockdown in spermatogonial stem cells (SSCs)and germ cell transplantation experiments, they show that SMN expression is regulated during spermatogenesis and SMN is involved in the maintenance of spermatogonia population.

The study is interesting and contributes to elucidate the role of SMN in male germ cells. However some points require other details and I think that manuscript would benefit from some additional explanation and modifications and in refinement of the language.

Specific suggestions:

Results section

In this study the authors isolate male germ cells at different stages of differentiation by FACS, on the basis of THY1 expression. Spermatogonia population is highly heterogeneous and includes undifferentiated cells (Plzf- positive) and differentiating spermatogonia (Kit-positive cells). A better characterization of these populations should be performed on the groups of cells considered in this study.

Fig. 1    

Panel C: Histograms show the results of qReal-time PCR. This is not indicated in the manuscript. Are the results reported as relative expression or fold induction? This is not clear.

Analysis of markers Acr1, Scp3 correlates with the stages of differentiation of isolated germ cells while expression of Plzf , a spermatogonia markers, is higher in 2wFSCHI (spermatocytes) than in 2wTHY+ (spermatogonia). This is not in accordance.

As for Smn1 expression, the qRealtime PCR results reported in the histogram do not correlate with the IF images of testis cross sections. In particular, the histogram shows that the expression of Smn1 in spermatogonia is higher than that in spermatocytes in 2weeks testis while in IF the strongest staining is detected in spermatocytes in the testis at the same age.

IF of testis cross sections from 8 weeks is not sufficiently clear. The authors should provide an image in which a complete cross section of seminipherous tubules can be observed.

Panel D: IF experiments show that in the seminipherous tubules, spermatocytes express the higher SMN protein level. However, the authors focused on spermatogonia stem cells. A

Fig. 2

Panels D-E-F: How many sections of each Ctrl and Sma-like mice have been analysed? It should be indicated in the figure legend. I think that almost 100 seminipherous tubules/animal randomly chosen in at least 3 non serial testis sections per animal, should be counted.

Fig. 3

Knockdown of Smn1 in magnetic-activated cell sorting (MACS) isolated spermatogonial stem cells

159 (SSCs) of C57/BL6 mice was performed to evaluate the effects of SMN depletion. The age of the mice from which SSC were isolated to perform Smn knockdown should be indicated.

Panel B reports the histogram of Smn and Plzf protein levels. Are these histograms the densitometric analysis of western blot in panel A as indicated in the legend? In the text (lane 162) panel B is reported as immunofluorescence analysis. Please check this point in the text and in the corresponding figure legend.

Panel B: The difference (significant or not) in the THY- cells between shLacZ and shSMN#1 and shSMN#2 should be indicated and discussed.

Fig. 4

Panel B: In the text they say: “The IHC staining (Fig. 4a) against R-loop structural marker, S9.6, indicated that SMA-like mice had modest number of S9.6-positive cells in seminiferous tubules (Figure 4b, upper panel). However, significantly higher number of S9.6-positive cells appeared in testis of SMA-like mice (Figure 4b, lower panel)”. It is not clear in the figure legend and in the text the difference between the upper and lower panels.

Round 2

Reviewer 2 Report

The suggested changes have been done and manuscript has been improved.